# Polish Paralympic Sports in the Opinion of Athletes and Coaches in Retrospective Studies

**DOI:** 10.3390/ijerph16244927

**Published:** 2019-12-05

**Authors:** Joanna Sobiecka, Ryszard Plinta, Marta Kądziołka, Wojciech Gawroński, Paweł Kruszelnicki, Anna Zwierzchowska

**Affiliations:** 1Faculty of Motor Rehabilitation, University of Physical Education, 31-571 Krakow, Poland; j.w.sobiecka@interia.pl (J.S.); martadelaf@wp.pl (M.K.); 2Department of Adapted Physical Activity and Sport, Chair of Physiotherapy, School of Health Sciences in Katowice, Medical University of Silesia, 40-752 Katowice, Poland; rplinta@sum.edu.pl; 3Department of Internal Medicine and Gerontology, Medical College, Jagiellonian University, 31-531 Krakow, Poland; w.gawronski@medicinasportiva.pl; 4Department of University School Informatization, University of Physical Education, 31-571 Krakow, Poland; pawel@kruszelnicki.eu; 5The Jerzy Kukuczka Academy of Physical Education, Institute of Sport Science, 40-066 Katowice, Poland

**Keywords:** Paralympic sport, limitations, training process, organizational structures

## Abstract

The study aimed to identify the limitations observed in Polish Paralympic sport depending on the environment in which athletes train on a daily basis. The study included 581 persons divided into two basic groups. The first group consists of athletes (n = 324) and coaches (n = 88) appointed to the national team by associations and unions providing sports training exclusively for athletes with disabilities. The second group consisted of athletes with disabilities (n = 146) and their coaches (n = 23), who work in national sports associations working for both able-bodied and disabled people. The study used the diagnostic survey method with a questionnaire developed by Sobiecka. The difficulties indicated by the respondents referred to various aspects related to the activity in professional sport. Particularly emphasised difficulties were related to organizational and financial limitations as well as the management and coaching staff. At the same time, it was demonstrated that the environment was a differentiating factor between the studied groups of athletes and coaches.

## 1. Introduction

It is well known that the successes of Paralympians are influenced not only by the work of the athletes themselves, but also by the coaching staff and the standards of conditions in the period of preparation for the Paralympic Games [1]. Comparative research of scientists from different countries involved in the sports achievements of Paralympians can provide valuable information in this respect [2]. An important role is also played by regular training in clubs (in addition to national team training camps) where disabled athletes practice their sports on a daily basis. Therefore, both systemic and legislative solutions are important, which in consequence requires the creation of appropriate conditions in various areas of functioning of athletes with disabilities. These include the organization of training and sporting events [3], coaching staff working with athletes [4,5], including psychologists [6], sports infrastructure [7], equipment and personal orthopaedic sports supplies [8,9,10]. Furthermore, in the opinion of Blauwet and Iezzoni, Rimmer et al., and Sahin and Lexell [11,12,13], it is important to ensure adequate funding and thus adequate marketing for sport for people with disabilities [9,14,15]. Blauwet and Iezzoni, Wedgwood and Calvente Rejón [11,16,17] emphasised the need for understanding and acceptance of sport for people with disabilities, especially in the environment of athletes with disabilities. Gawroński et al. and Simon and Ward highlighted the importance of care and specialized medical staff in the process of training of athletes with disabilities [18,19]. Furthermore, diagnosis of injuries, including those dangerous to health, and appropriate treatment to ensure the return to sport are also essential [20]. Little attention has also been paid to research into adequate nutrition and supplementation in sports for people with disabilities, including Paralympic sports [21]. Sport is very important in promoting healthy lifestyles and preventing health problems [22].

The above individual suggestions have not been reflected in the literature of comprehensive scientific research that identified the actual conditions in which athletes with disabilities train every day in Poland. World literature also draws attention to the lack of sufficient empirical research into the important components of training for people with disabilities at the highest sports skill levels and the barriers affecting the interest in and participation of children with disabilities in sport. This is confirmed by Dowling et al. in their study with a very broad literature review [23]. The first interests in this area were observed in Poland in 1998–1999. A study was conducted among athletes participating in the Paralympic Games in the period of 1992–1998, and was continued in the following years [24]. They showed that during sporting careers, athletes have had to overcome a number of difficulties. In each period of research, they were mainly related to the lack of adequate funding for sports activities of people with disabilities. They also mentioned the lack of support from the professional coaching staff, and insufficient training facilities, highly specialized sports equipment or personal sports orthopaedic supplies. What was worrying, however, was the fact that the proportion of athletes with disabilities affected by inadequate prevention and medical care, and the lack understanding in associations and societies in which they were affiliated, was growing steadily. With the above aspects, the aim of the research was to identify and stratify the limitations that occur in Polish Paralympic sport depending on the environment in which athletes train on a daily basis. It was assumed that the environment in which disabled athletes function is significant for their success in sport. 

## 2. Material and Methods

### 2.1. Material

The survey was conducted between November 2015 and December 2016. The respondents included members of all Polish associations and associations acting exclusively for the promotion of Paralympic sport. Associations and unions provide sports training only for players with disabilities, while sports unions operate in one specific sport, both for players with disabilities and those able-bodied.

In the period of the survey, the number of disabled people in Poland appointed to the Polish national team was n = 483, of whom n = 470 responded. This accounted for 97.3% of the general population of disabled members of the national team. Furthermore, the number of coaches appointed at that time to the Polish national team was n = 113, of whom n = 111 joined the survey, which translated into 97.9% of the respective general population of coaches of people with disabilities. Therefore, the study included 581 persons in total, divided into two basic groups (Group 1 and Group 2).

Group 1 consisted of participants appointed to the national Paralympic team by all-Polish associations and unions providing sports training only for athletes with disabilities. 

This group was comprised of 324 individuals (70 women, 254 men) with locomotor or visual impairments and 88 coaches (10 women, 78 men). 

Group 2 was also composed also of members of the national Paralympic team, but they were nominated by all-Polish Sports Associations operating within specific sports for both able-bodied and disabled athletes. This group consisted of 146 athletes (28 women, 118 men) with locomotor disorders and 23 coaches (4 women, 19 men)—Figure 1. 

### 2.2. Methods

The method of retrospective studies was applied, with the use of a diagnostic survey by means of a questionnaire designed by Sobiecka [25]. The questionnaire was modified for the purposes of the research and was entitled “The process of integration of Polish sports communities” [26]. The research procedure consisted of four stages (Figure 2). 

In the first stage, based on the documentation of the Ministry of Sport and Tourism [27], direct contact was established with the chairpeople of the boards of all national organizations (19), in which coaching in disabled sports occurred in 2016, including Paralympic sports. Unions and associations operating exclusively for athletes with visual and locomotor disabilities were included. During the individual interviews, chairpeople of these organizations (and in the further stages of the research, also the coaches and athletes) were informed about the purpose of the research, the general research problems were presented, and the chairmen were asked for consent to conduct the research. Next, based on the documentation available in the above organizations, a personal list of athletes and their coaches appointed to the national team in particular sports was prepared.

In the second stage, contact was established with all coaches of the Paralympian teams in Poland, with whom direct interviews were conducted. Furthermore, taking into account the coaching documentation, the data on the national team members for 2016 was verified and updated.

In the third stage, the survey was conducted after obtaining the approval of the competitors. The survey was carried out in Poland, during national competitions, camps, central consultations and training sessions in sports clubs. The respondents were athletes and coaches of the Polish Paralympic Team.

In the last stage of the research, the respondents’ opinions were extracted from the survey questionnaire, concerning only the limitations occurring in the Polish Paralympic sport. The data analysis was based on descriptive statistics, presenting the results for each group of respondents (players and coaches). The sports environment (unions and associations—6 organizations providing sports training only for athletes with disabilities), as well as Polish sports unions (8 organizations acting in one specific sport, both for able-bodied and disabled athletes) was also taken into account. 

The questionnaire for female and male athletes and the national Paralympian team consisted of the following parts: the introductory instruction, the main part and the respondent data. In the introductory instruction portion, the aim of the study was specified and the respondents were asked to emphasise the chosen answer and/or write their opinion. Furthermore, in the specified page of the questionnaire, the respondents could add comments or opinions that arose during its filling in the form. 

The main part of the survey questionnaire consisted of 11 items, which were mainly semi-open (7), while 3 of them were open and one was closed. They concerned the integration of the disabled sports with the sports community of able-bodied athletes, positive aspects and difficulties in the sport, as well as forms of support expected from national sports organizations, which would make it easier for players with disabilities to participate in the sport. There were also items concerning the organizational and substantive preparation of Polish sports unions to take over athletes with disabilities who practice sports, the differences in training between Polish sports unions and organizations dealing exclusively with sports for people with disabilities, and the ability of athletes to comply with the requirements of the training process specified by Polish sports unions. The questions also contain information on the rights and obligations of the athletes appointed to the Paralympic national team and coaches who athletes with disabilities would like to work with.

The next part of the survey questionnaire (respondent data) contained 16 questions. They concerned the sociodemographic characteristics of the athletes studied in the period of conducting the research. The questions covered the following aspects: disability, sporting activity, material conditions and other characteristics, concerning age, sex, marital status and educational structure of the athletes surveyed (Appendix A); 

The survey questionnaire for the coaches of the Paralympic national team also consisted of three parts. The differences were in its main part, which included fewer items (8 questions). The questions were semi-open (4), open (3) and closed (1). Comparing it with the questionnaire for athletes of national Paralympic team, there were no questions about the rights and obligations of players appointed to the national team and about coaches who athletes with disabilities would like to work with. The respondent data part included 13 questions concerning sociodemographic features of coaches at the time of the survey. They included: sex, age, marital status, education structure, coach’s certificates and coaching activity (Appendix B). In order to verify the significance of the correlations for the problems from individual areas between the above mentioned sports organizations, we used tests for two structure indicators. Furthermore, chi-squared tests were used to demonstrate differences in the choice of the dominant characteristic in a given problem area.

It should be noted that ethical principles formulated in the International Guidelines for the International Ethical Guidelines for Biomedical Research Involving Human Subjects (developed by the Council for International Organizations of Medical Sciences in collaboration with the World Health Organization, adopted in 1982 and revised in 1993 and 2002) were respected at all stages of research. (International ethical guidelines for biomedical research involving human subjects. Geneva: Council for International Organizations of Medical Sciences, 2002 (access on 17 January 2018). 

## 3. Results

About 98% of athletes and 99% of coaches from the national team reported limitations occurring in the Paralympic sport in Poland. The difficulties identified by the respondents covered seven major areas. The difficulties were classified according to their prevalence, with the respondents indicating: (1) lack of regular financial support for the development of sport for people with disabilities, (2) low level of promotional and marketing activities in sport for people with disabilities, (3) difficulties and barriers in the organization of training and competitions for people with disabilities, 4) difficult access to professional coaching staff, (5) low awareness in the understanding of the limitations of disabled sports, (6) limited access to sufficient training facilities, advanced sport-specific equipment and personal athletic orthopaedic supplies, (7) difficulties in access to medical care, psychologists, dietitians and physiologists.

According to the respondents, limitations in Polish sports organizations providing training in Paralympic sports are mainly related to financing (Table 1). Among other things, the lack or inadequate management of regular financial support for the development of sports was also highlighted. In the opinions, statistically significant differences appeared only for athletes from associations and unions and from Polish sports associations. Athletes appointed to the national team by Polish sports associations were significantly more likely to indicate the lack of regular financial support for the development of sports, while those who trained within associations and unions providing coaching only for athletes with disabilities pointed to inadequate financing of Paralympic sports (chi-2 = 78.46; df = 2; *p* < 0.05). 

Marketing in sport for disabled people is another significant area indicated by disabled athletes and their coaches and analysed in the study (Table 1). Paralympic athletes and their coaches mainly pointed to difficulties in finding sponsors, lack of promotion and fundamental work in terms of cooperation between clubs, sports associations and educational institutions, the environment of parents and legal guardians, and consequently, recruitment of children and youth with disabilities. The opinions of the respondents were significantly different both among coaches (chi-2 = 8.15; df = 2; *p* < 0.05) and players (chi-2 = 13.73; df = 2; *p* < 0.05), but it did not depend on the sports environment in which they were training.

Table 2 presents an analysis of the data concerning the area related to the organization of training and sports competitions in Polish Paralympic sport. The majority of athletes and coaches agreed that the main limitation is the lack of systemic solutions, procedures of recruitment for sports among adults with disabilities. Furthermore, they indicated a small number of camps and sports competitions at different levels. Statistically significant differences in these statements were observed only among the players. These opinions were much more often indicated by athletes of national teams training in associations and unions than in Polish sports associations (test chi-2 = 7.89; df = 2; *p* < 0.05).

Coaching staff, referees, volunteers and assistants supporting players with disabilities during training sessions and sports competitions is another aspect related to the area of organization of training and sports competitions in Polish Paralympic sport (Table 2). Limited access to the coaching staff, insufficient number of referees in Paralympic sports are opinions that were statistically significantly differentiated only in the group of athletes. Athletes appointed to the Polish National Team by associations and unions providing training only for athletes with disabilities indicated them much more often than those training in Polish sports associations (test chi-2 = 5.56; df = 1; *p* < 0.05).

Understanding the limitations of disabled sports is the only area which statistically significantly differentiated the opinions of both athletes and coaches with regard to the sports environment from which they came from (Table 3). Players and coaches were critical of the level of knowledge about disability and indicated the lack of understanding and acceptance of players with disabilities in Polish sports associations. They pointed to the underestimation of their sporting achievements and the lack of proper cooperation and communication in relations between the union, club, coaches and athletes. These opinions were significantly more frequently presented by the athletes surveyed coming from the environment providing sports training only for people with disabilities than from Polish sports associations (chi-2 = 4.74; df = 1; *p* < 0.05). 

Furthermore, Table 3 includes the opinions of respondents related to the organizational structure of Polish sport for disabled people and competencies of people working for the associations. Players from Polish sports associations were much more likely to indicate the small number of clubs and sports sections for people with disabilities, whereas those from associations and unions pointed more often to ineffective structures of Polish sports for the disabled (chi-2 = 6.11; df = 2; *p* < 0.05).

The availability of highly specialised sports equipment and personal orthopaedic supplies is one of the complementary areas concerning the organization of sports training (Table 4). Both athletes and their coaches pointed to these limitations, and statistically significant differences in these statements were recorded only among players due to the environment of their participation in sports training. 

Other obstacles that were pointed out by the respondents were the lack of adequate training and hotel facilities and limited access to sports facilities, taking into account the needs of people with disabilities (Table 4). They were especially emphasized by coaches and those who provided training in associations and unions acting only for the benefit of disabled sports. There were no statistically significant differences in the opinions of athletes. 

The results presented in Table 5 show other difficulties, e.g., those related to medical care and access to psychologists, dietitians and physiologists. The most frequent limitation mentioned by the respondents was the lack of regular and professional prevention and medical care. Statements about the difficulties mentioned above turned out to be significantly differentiated only in the athletes’ environment. No statistically significant differences were found in the opinions of coaches. 

## 4. Discussion

The above results revealed the limitations that occur in Polish Paralympic sport and cover many areas of the activities related to this sport. It should be stressed, however, that the presented results are among the first to be indicated by so many people directly involved in professional sports for the disabled. The authors of the paper did not find similar studies in international literature. Therefore, their findings cannot be referred to the results of other national representations of Paralympic sports. It should be remembered, however, that the dominating obstacle of the lack of adequate funding has been reported by Polish Paralympic athletes in studies published in 1992–1998 (about 46%), 2000–2002 (about 33%) and 2004–2006 (about 32%) [24], and by the 2016 National Team athletes competing in Paralympic sports. They emphasized, among other things, unfair criteria for awarding scholarships, the lack of awarding athletes for setting records of Paralympic Games, world or European Championships, and too-low stipends [26]. Furthermore, the lack of regular financial support for the development of sports turned out to be the most frequently reported limitation, both by the athletes and their coaches. This manifested itself in the lack of or incomplete reimbursement of the competitors’ costs of stay at the camps and competitions, fees for the competition license in a given sport or insurance for international trips [26]. Another difficulty was the inadequate financial policy, which was noticed mainly by the coaches surveyed. Most often, it concerned the allocation of funds against their intended use and the inappropriate division of funds between sports and even clubs within a given association.

With the above-mentioned limitations of the Polish Paralympic sport, and the comments of Blauwet and Iezzoni [11] and Sahlin and Lexell [13] that financial difficulties may limit the opportunities of people with disabilities to participate in sporting activity, and sometimes even exclude them, it can be assumed that in the future, the area of this activity in Poland may become inaccessible to those willing to practice sport, not only in the form of rehabilitation but, above all, as professional work. In this context, it is worth quoting the opinions of Rimmer et al. [12]. The authors found that the participation of people with disabilities in sporting activity is costly, thus being the barrier to participation, even in developed countries. It should be noted, however, that in Poland, financing of sporting activity for the disabled is mainly based on funds from the state budget. Therefore, one should consider whether the problem of financing sporting activity arises only from a lack of resources or perhaps from inefficient spending.

Another limitation in disabled sports is the lack of interest of sponsors or business organizations. It is well known that the media, especially television and the Internet, through the globalization of sports events, encourage the involvement of sponsors [28]. Unfortunately, despite the participation of Polish competitors in the top level sporting events (Paralympic Games, world or European championships), finding sponsors is very difficult. A similar problem is observed in other European countries, including Spain [29]. Therefore, social marketing should be used to a greater extent to promote sport, as mentioned by Cottingham et al. [14] and Curran and Hirons [9], especially as associations and unions dealing exclusively with disabled athletes are non-profit organisations. Furthermore, they have serious problems with financing the promotion of disabled sports, which has been reported by representatives of the national team of particular sports [26]. Therefore, it seems that there is no awareness in Poland of the benefits of socially engaged marketing, although it can improve living conditions, promote a brand, build relationships with consumers and offer many other benefits [15].

Paralympic sport experiences its internal problems with the organization of training. This applies to both clubs and the Polish national team. Respondents from all groups pointed out the lack of reliable information on the possibilities of practising sport by people with various dysfunctions. Despite significant progress in showing and promoting Paralympic sports compared to the early 1990s, no effectively organized recruitment system for people with disabilities (including young people) to take part in sport was found in Poland. At the level of sports clubs, the training process is not subject to any coordination. The authors’ observations and experience show that occasional trainings are often limited to the participation of coaches of the national team, and often concern many sports. Therefore, the topics discussed during such trainings are interdisciplinary.

The coaching staff, especially those cooperating with athletes and referees of Paralympic sports on a daily basis, was subjected to criticism by the respondents. The main comments concerned the lack or limited access to professionals. It seems evident that one of the reasons for this is the lack of funds for the training of qualified coaching staff to suit the needs of disabled sports [26]. Another difficulty is related to unattractive salaries and the lack of employment opportunities for coaches under employment contracts in sports clubs. This was confirmed by people working for disabled athletes in Poland. They stated that too little money was spent on coaches’ positions [26]. It should be mentioned, however, that for most coaches, this is their additional work. With this in mind, and taking into account the opinions of Bastos et al. [4] and Martin and Whalen [5] about the significant role of both female and male coaches in sports professionalization and achievement of the maximum potential of elite players with disabilities, it can be concluded that the further development of Polish professional sport raises many concerns.

Another limitation, which was expressed in the statements of the athletes surveyed and their coaches, occurring in Polish Paralympic sport, was the understanding of the limitations of disabled sports. In this respect, Blauwet and Iezzoni [11] and Wedgwood [16] even emphasized the need to understand this activity in the sports environment of people with disabilities. Despite the rapid development of the Paralympic movement, athletes still face many barriers to participation and inadequate presentation of their image in sport at the highest level [30].

Another limitation mentioned by the respondents was the small number of clubs and sports sections. They operate within the structures of Polish sport and are usually based in big cities. This is particularly noticeable in the Polish Sports Association for the Disabled “Start”, which conducts training in ten summer and five winter Paralympic sports [26]. However, one cannot ignore the fact that the attempt to take over organizational patterns from some countries, consisting in the full integration of able-bodied and disabled athletes into one organization [31] has not brought the expected results in Poland at present. Polish sports associations take over certain sports of disabled people only at the level of national teams, whereas training in clubs continues in associations and unions acting exclusively for the benefit of people with disabilities. This causes some problems, for example, with coordination in training and organizational matters [26].

In the opinion of the respondents, there are also barriers concerning access to high-quality specialized sports equipment, including personal sports orthopaedic supplies. It would be reasonable to assume that they are caused by different aspects. The equipment must be selected individually for each competitor. If an athlete is excluded from training and competitions for health reasons or withdrawal from practising a sport, there is little chance of the equipment being used by another athlete. At the same time, it should be remembered that the purchase of equipment consumes high financial resources, which were mentioned by both coaches and athletes [26]. Consequently, it can be assumed that the presented difficulties become a serious factor that limits the achievement of high sports performance among disabled players taking part in professional sports in Poland. This was also emphasized by Jaarsma et al. [7] and Kean et al. [10].

The coaches from both environments (associations and unions for the disabled and Polish sports associations) also noticed limited access to appropriate training facilities. These include, for example, good location, accessibility for people with various disabilities or convenient training hours [26]. As can be seen, this was not always caused by insufficient financial resources for renting sports facilities, even though Sobiecka reported on these limitations [26]. At this point, the opinion of Campbell and Jones [32] becomes important: that the lack of accessible sports facilities for people with disabilities is one of the main sources of stress for elite athletes.

A serious and negative phenomenon affecting the conditions of practising sport by people with disabilities is the lack of prevention in the form of regular medical check-ups in the field of sports medicine and access to proper medical care. This limitation is particularly emphasized by the athletes surveyed and coaches from organizations acting exclusively for the benefit of disabled sports. It has also been reported by the Polish Paralympians in 1992–1998 (ca. 12%), 2000–2002 (ca. 17%), 2004–2006 (ca. 17%) [24]. Positive changes in this respect could be observed only in the period of preparation for the London 2012 Paralympic Games [1]. This was probably due to the implementation in the Polish national team, in the form of enforcing before the Games, preventive multispecialist examinations in the field of sports medicine. Such a procedure was recognized not only by the Paralympic Games, but also proved to be effective in reducing the incidence of musculoskeletal injuries and illnesses during the Paralympic Games [18]. Therefore, it may be suggested that similar medical measures should be introduced at a club level. Undoubtedly, observance of regular preventive medical check-ups in the field of sports medicine, completed with the certificate of health status, will ensure that athletes are able to participate safely in sport, both during training and competitions. The development of modern civilization that brings awareness of the society about disabilities requires conducting educational and promotional activities, especially among parents and legal guardians, concerning the possibility of practising sport as an antidote to disability. These measures require media support, which, unfortunately, is also lacking in the area of professional sport for disabled people, both in Poland and in other countries. Often, high-level competitions such as the Paralympic Games, are being pushed into the background of television broadcasts and replaced by commercial broadcasts of less important competitions [29].

## 5. Conclusions 

The difficulties indicated by the respondents referred to various areas related to their activity in professional sport. The sports environment turned out to be a factor significantly differentiating mainly the players in response to the questions concerning the limitations of the Polish Paralympic sport. Negative opinions were significantly more frequent, as indicated by respondents from the national team appointed by associations and unions providing sports training exclusively for people with disabilities in comparison to Polish sports associations. 

Regardless of the environment in which players with disabilities train every day, the burden of limitations primarily concerns financial and organizational matters or issues related to the coaching and support staff. 

## Figures and Tables

**Figure 1 ijerph-16-04927-f001:**
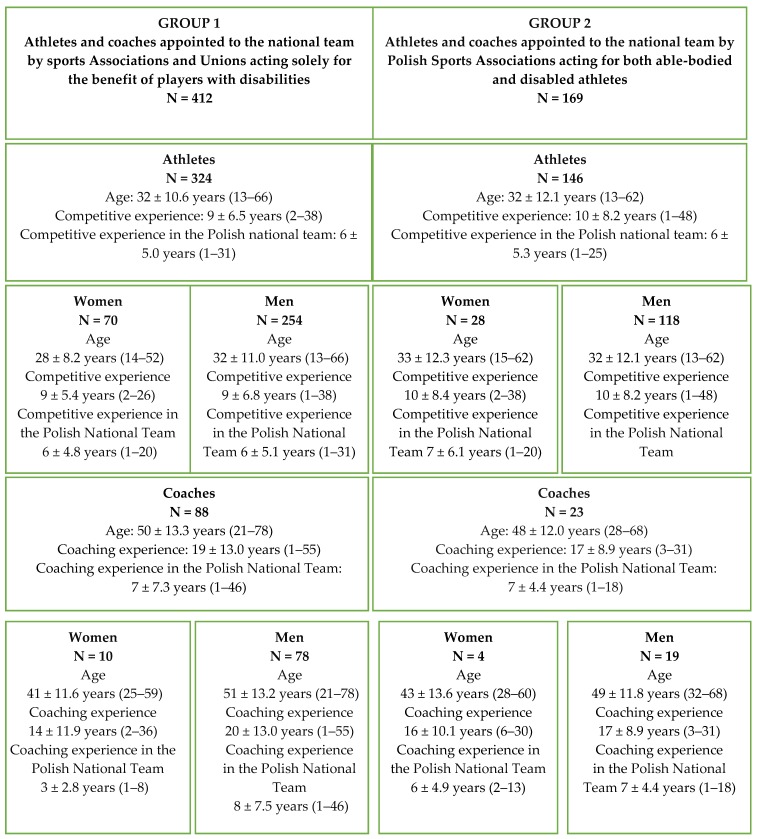
Characteristics of the athletes with disabilities and their coaches by groups.

**Figure 2 ijerph-16-04927-f002:**
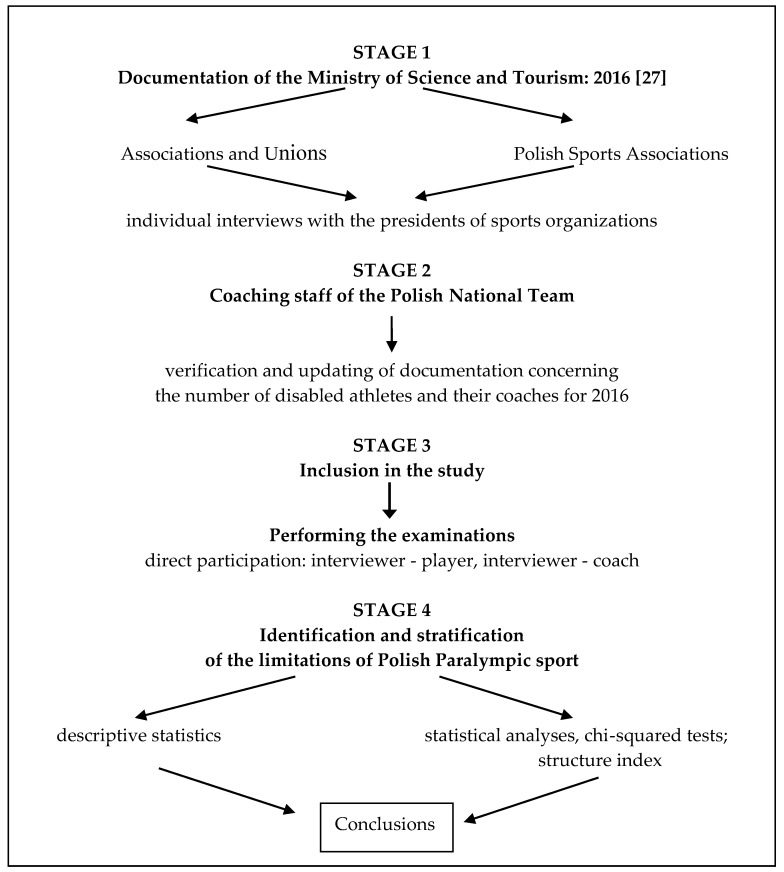
The procedure of retrospective research conducted with the questionnaire [25,26].

**Table 1 ijerph-16-04927-t001:** Main limitations in Polish Paralympic sports relating to financial support and marketing in disabled sports (according to the studied athletes and coaches of the national team) *.

Limitations in Polish Paralympic Sports	Study Population					
	Associations and Unions		Polish Sports Associations		Total	
	Athletes n = 324	Coaches n = 88	Athletes n = 146	Coaches n = 23	Athletes n = 470	Coaches n = 111
	%	%	%	%	%	%
**FINANCIAL SUPPORT **** (*Athletes from Associations and Unions vs. athletes from Polish Sports* Associations *p = 0.000)*						
Lack of regular financial support for the development of sports	41.4	79.4	69.2	78.3	50.0	79.3
Inadequate or no financial support	64.8	18.1	10.9	17.4	48.1	18.0
Ineffective financial policy	6.2	64.8	4.1	56.5	5.5	63.1
Complicated procedures for obtaining financial support or reimbursement	1.2	10.2	0.7	0	1.1	8.1
Lack of rules concerning remuneration of guides and pilots of blind athletes	0.9	1.1	1.4	0	1.1	0.9
**MARKETING IN DISABLED SPORTS**						
Difficulties in obtaining sponsors	14.5	22.7	24.0	17.4	17.4	21.6
Lack of promotion; no recruitment of disabled children and teenagers; no cooperation of clubs with schools, occupational therapy workshops and training-education centres; lack of informative awareness-raising events for parents and legal custodians of disabled children and teenagers to promote disabled sports; lack of junior and youth national teams	12.3	22.7	10.3	47.8	11.7	27.9
Lack of knowledge, interest and understanding of disabled sports in the society; lack of appreciation for achievements in disabled sports	13.0	22.7	5.5	8.7	10.6	19.8
A small number of athletes training and competing is some sports and classes	5.2	9.1	2.0	0	4.3	7.2

* The total does not equal 100% as the respondents were free to express more than one opinion; ** level of statistical significance *p* < 0.05.

**Table 2 ijerph-16-04927-t002:** Limitations in Polish Paralympic sports relating to the organization of training and competitions, as well as to the training and support personnel (according to the studied athletes and coaches of the national team) *.

Limitations in Polish Paralympic Sports	Study Population					
	Associations and Unions		Polish Sports Associations		Total	
	Athletes n = 324	Coaches n = 88	Athletes n = 146	Coaches n = 23	Athletes n = 470	Coaches n = 111
	%	%	%	%	%	%
**ORGANIZATION OF TRAINING AND COMPETITIONS **** *(Athletes from Associations and Unions vs. athletes from Polish Sports* Associations *p = 0.02)*						
Lack of information; lack of a recruitment system of disabled athletes	49.1	56.8	27.4	60.8	42.3	57.6
A small number of competitions and training camps organised at different levels	15.1	9.1	6.2	4.3	12.3	8.1
Inconvenient commuting for regular training, training camps and competitions	7.1	2.3	13.0	13.0	8.9	4.5
A small number of athletes training and competing in some sports and competition classes	5.3	0	0.7	0	3.8	0
Different performance levels among athletes within sections or clubs, limiting the possibility to obtain good results	1.9	2.3	2.7	0	2.1	1.8
Unclear rules on athlete selection for the national team (including Paralympic games)	1.2	0	1.4	4.3	1.3	0.9
Unprofessional organization of sports competitions	0	1.1	0.7	0	0.2	0.9
Lack of a standardised training system for disabled athletes in associations and Polish Sports Associations	0	1.1	0	8.7	0	2.7
**COACHING AND SUPPORT PERSONNEL **** *(Athletes from Associations and Unions vs. athletes from Polish Sports* *Associations p = 0.002)*						
No or limited access to professional personnel engaged in work with disabled athletes; a small number of competent coaches and Paralympic sports officials	23.8	36.4	7.5	30.4	18.7	35.1
Lack of or insufficient number of volunteers and assistants helping during training and competitions	4.0	5.7	3.4	4.3	3.8	5.4

* The total does not equal 100% as the respondents were free to express more than one opinion; ** level of statistical significance *p* < 0.05.

**Table 3 ijerph-16-04927-t003:** Main limitations in Polish Paralympic sports relating to the understanding of the question of disabled sports, including women’s sport, as well as the organizational structure (according to the studied athletes and coaches of the national team) *.

Limitations in Polish Paralympic Sports	Study Population					
	Associations and Unions		Polish Sports Associations		Total	
	Athletes n = 324	Coaches n = 88	Athletes n = 146	Coaches n = 23	Athletes n = 470	Coaches n = 111
	%	%	%	%	%	%
**UNDERSTANDING DISABLED SPORTS (INCLUDING WOMEN’S SPORT) **** (*Athletes from Associations and Unions vs. athletes from Polish Sports**Associations p = 0.001; coaches from Associations and Unions vs. coaches from Polish Sports**Associations p = 0.011)*						
Lack of knowledge, understanding and acceptance of disabled athletes in Polish Sport Associations; lack of appreciation of their sports achievements; lack of correct cooperation and communication between societies, clubs, coaches and athletes	32.6	21.6	8.2	30.4	25.1	23.4
Lack of endorsement from individuals working in disabled sports institutions	8.6	14.8	6.2	39.1	7.7	19.8
Low participation of women in sport	0.3	0	0.7	0	0.4	0
**ORGANIZATIONAL STRUCTURE AND COMPETENCE OF INDIVIDUALS** **INVOLVED IN DISABLED SPORTS**						
A small number of disabled sports clubs and sections	5.2	11.4	10.9	4.3	7.0	9.9
Ineffective structures within Polish disabled sports	6.5	10.2	5.5	4.3	6.2	9.0
No cooperation between clubs, associations, Polish Sports Association for the Disabled "Start and Polish unions	5.6	7.9	1.4	17.4	4.2	9.9
No coordinators supervising the cooperation between the management, coaches and disabled athletes in Polish Sports Associations	0	2.3	1.4	0	0.4	1.8
No representatives of some Polish national societies and associations active in the management board of the Polish Paralympic Committee	0	3.4	0	8.7	0	4.5

* The total does not equal 100% as the respondents were free to express more than one opinion; ** level of statistical significance *p* < 0.05.

**Table 4 ijerph-16-04927-t004:** Main limitations in Polish Paralympic sports relating to access to sports equipment, personal orthopaedic supply and the training site (according to the studied athletes and coaches of the national team) *.

Limitations in Polish Paralympic Sports	Study Population					
	Associations and Unions		Polish Sports Associations		Total	
	Athletes n = 324	Coaches n = 88	Athletes n = 146	Coaches n = 23	Athletes n = 470	Coaches n = 111
	%	%	%	%	%	%
**ACCESS TO SPORTS EQUIPMENT AND PERSONAL ORTHOPAEDIC SUPPLY FOR SPORTS **** *(Athletes from Associations and Unions vs. athletes from Polish Sports* *Associations p = 0.001)*						
Limited access to highly specialised sports equipment	20.4	20.4	15.1	26.1	18.7	21.6
Limited access to highly specialised personal orthopaedic supply for sports	20.1	17.0	0	26.1	13.8	18.9
**TRAINING SITE **** *(Coaches fromAssociations and Unions vs. coaches from Polish Sports* *Associationsp = 0.047)*						
Lack of appropriately adapted training sites and accommodation, limited access to sports structures for the disabled	18.2	52.3	19.2	26.1	18.5	46.8

* The total does not equal 100% as the respondents were free to express more than one opinion; ** level of statistical significance *p* < 0.05.

**Table 5 ijerph-16-04927-t005:** Main limitations in Polish Paralympic sports relating to medical care and support of psychologists, dietitians and physiologists (according to the studied athletes and coaches of the national team) *.

Limitations in Polish Paralympic Sports	Study Population					
	Associations and Unions		Polish Sports Associations		Total	
	Athletes n = 324	Coaches n = 88	Athletes n = 146	Coaches n = 23	Athletes n = 470	Coaches n = 111
	%	%	%	%	%	%
**MEDICAL CARE AND SUPPORT OF PSYCHOLOGISTS, DIETITIANS AND PHYSIOLOGISTS **** *(Athletes from Associations and Unions vs. athletes from Polish Sports* *Associations p = 0.004)*						
Lack of systematic prophylaxis and appropriate regular medical care (physicians, physiotherapists, masseurs)	13.0	21.6	3.4	13.0	10.0	19.8
No support from a psychologist	5.6	6.8	3.4	17.4	4.9	9.0
No support from a dietitian	1.9	2.3	0.7	0	1.5	1.8
No support from a physiologist	0.6	1.1	0	0	0.4	0.9

* The total does not equal 100% as the respondents were free to express more than one opinion; ** level of statistical significance *p* < 0.05.

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
