# Peer review of "Polish Paralympic Sports in the Opinion of Athletes and Coaches in Retrospective Studies"

_ijerph, 2019, doi:10.3390/ijerph16244927_

Round 1
Reviewer 1 Report
The Introduction part.
This part would be enhanced by the highlighting of the difference between professional and amateur athletes in sport for disable people. Also would nice to be found a broader discussion of environmental influences on disability athletes' preparation and rezults, as well on their well-beeing.
Authors could highlight the object of the research and the research problem itself.
The Material and Methods part.
This part lacks clarity.
It is not clear why both athletes and coaches were chosen as subjects of the reseach.
It is not clear why only 14 sports associations and societies were inducded in the study. What is the different between sports associations and societies.
Most of the uncertainty is caused by Figure 2 (The procedure of retrospective research conducted with the questionnaire) presented by the authors.
There is mising data from stages 1, 2 and 3. Was the goal of stage 1 just to get “consent to conduct the reseach”?. If so for what purpose the individual interviews with the presidents of sports organizations was conducted. And how the interview data have been processed. The same question about STAGE 3.
It is also not explained the research strategy was chosen for the main study – is it qualitative or quantitative? The research instrument is not fully explained. The validity and reliability of the questionnaire are not discussed.
In the discussion section, the authors interpret some results based on a non-professional sport approach (sport for all disable people). Therefore, it remains unclear what is the difference between professional and amateur disables athletes.
Author Response
I am submitting a revised version of my paper entitled „ Polish Paralympic sports in the opinion of athletes and coaches in retrospective studies”. I gave each of the points raised by the reviewers careful consideration and made all changes needed to fulfill their intention as well as formal requirements. The corrections and amendments are marked red in the text.
Point 1The Introduction part.This part would be enhanced by the highlighting of the difference between professional and amateur athletes in sport for disable people. Also would nice to be found a broader discussion of environmental influences on disability athletes' preparation and rezults, as well on their well-beeing.
Response 1In accordance with the suggestion of the reviewers and the editor, the most recent references have been added to improve the argumentation of the introduction to the problems discussed in the paper. They have been used in the Discussion section. The following reference items have been added:
Dowlinga M., Brown b , Leggc D, Beacom: A Living with imperfect comparisons: The challenges and limitations of comparative paralympic sport policy research Sport Management Review 21 (2018) 101–113 Castro-Sánchez, M., Zurita-Ortega, F., Chacón-Cuberos, R., López-Gutiérrez, C. J., & Zafra-Santos, E. (2018). Emotional Intelligence, Motivational Climate and Levels of Anxiety in Athletes from Different Categories of Sports: Analysis through Structural Equations. International journal of environmental research and public health, 15(5), 894. doi:10.3390/ijerph15050894 Calvente Rejón, I. (2016). Tratamiento de los Juegos Paralímpicos por parte de los medios de comunicación: Ambición y coraje en los Paralímpicos de Río 2016. Kirk B1, Pugh JN2, Cousins R3, Phillips SM Concussion in University Level Sport: Knowledge and Awareness of Athletes and Coaches.Sports (Basel).2018 Sep 20;6(4). pii: E102. doi: 10.3390/sports6040102. Madden, R. F., Shearer, J., & Parnell, J. A. (2017). Evaluation of Dietary Intakes and Supplement Use in Paralympic Athletes. Nutrients, 9(11), 1266. doi:10.3390/nu9111266 Kwok Ng & Kasper Mäkelä & Jari Parkkari & Lasse Kannas & Tommi Vasankari & Olli J. Heinonen & Kai Savonen & Lauri Alanko & Raija Korpelainen & Harri Selänne & Jari Villberg & Sami Kokko, 2017. "Coaches’ Health Promotion Activity and Substance Use in Youth Sports," Societies, MDPI, Open Access Journal, vol. 7(2), pages 1-11, April Shields, N., Synnot, A. J., & Barr, M. (2012). Perceived barriers and facilitators to physical activity for children with disability: a systematic review. Br J Sports Med, 46(14), 989-997. Solves, J., Sánchez, S., & Rius, I. (2018). The prince and the pauper: Journalistic culture and Paralympic Games in the Spanish print press. Journalism, 19(12), 1713-1729
Point 2Authors could highlight the object of the research and the research problem itself.
Response 2 According to the suggestion, a hypothesis was formulated for the purpose of the research
Point 3The Material and Methods part.This part lacks clarity
Response 3 - this part has been corrected with consideration for the acceptance for this chapter of the Reviewer no. 1
Point 4It is not clear why both athletes and coaches were chosen as subjects of the reseach. -
Response4 - the study group and the way it was selected for the study have been explained
Point5 It is not clear why only 14 sports associations and societies were inducded in the study. What is the different between sports associations and societies.
Response5 The differences between unions and associations have been explained in the Methodology sections (all unions and associations operating in Poland in 2016/17 were included in the survey
Point6 Most of the uncertainty is caused by Figure 2 (The procedure of retrospective research conducted with the questionnaire) presented by the authors.
There is mising data from stages 1, 2 and 3. Was the goal of stage 1 just to get “consent to conduct the reseach”?. If so for what purpose the individual interviews with the presidents of sports organizations was conducted. And how the interview data have been processed. The same question abopisane etapy badaÅ„ sout STAGE 3.
Response6 The stages of the study in Fig. 2 are consistent with the research. Stage I - obtaining consent, moving to stage II - verification and updating of the documentation obtained from the Ministry of Sport and Tourism concerning athletes and their coaches from the national team. It was at the third stage when the athletes and coaches underwent direct interviews with the interviewer, after prior consent obtained from them.
Point 7It is also not explained the research strategy was chosen for the main study – is it qualitative or quantitative? The research instrument is not fully explained. The validity and reliability of the questionnaire are not discussed.
Response 7 Validation of the tool was described in detail in the published study [18]. The present study used only selected aspects of the designated tool for the achievement of the purpose of the study.
Point 8 In the discussion section, the authors interpret some results based on a non-professional sport approach (sport for all disable people). Therefore, it remains unclear what is the difference between professional and amateur disables athletes.
Response8 The discussion is deliberately focused on Paralympic professional sport, because the publications in the aspect of sport as a means of rehabilitation for people with disabilities discuss the multitude of problems, limitations and barriers. In the opinion of the authors, this could only lead to misinterpretation of our results.
Reviewer 2 Report
In this manuscript, the authors examined “Polish Paralympic sports in the opinion of athletes and coaches in retrospective studies”. To this end, the authors used data from 581 people. As far as I know, the authors did a good job, especially, directing the data analysis. It seems that the results showed support for the objectives and the findings were discussed slightly. In general, I found that this is a well-written article that explores important ideas of relevance to the context of Polish Paralympic Sports. Therefore, I recommend some important revisions and taking into account some modest recommendations that could improve the document:
Line 33: Remove the word "problem", and replace it with a less ambiguous one.
Line 36-50: There are many influential aspects in Paralympic sport in a very superficial way. The authors express that there are no similar studies in Polish samples, however, they do not reflect similar studies in other countries. It would be necessary and interesting to include studies identifying the main difficulties encountered in sports for people with disabilities, even in different countries. In relation to the comments, I consider that the bibliography is poorly updated. I recommend to the authors the detailed search of more current studies related to the subject of study.
Figure 1: Congratulate the authors for the complete description of the sample.
Line 180: I suggest putting a title that refers to the paragraph that follows, and removing "The process of integration of Polish sports communities" along with the bibliographic reference, because it is confusing if it refers to a title or not.
Methods: There is no information about the modified questionnaire used, nor the type of questions asked, nor the type of answers. Were they all open questions? Were there closed questions? Were there open and / or closed answers? Likert responses? I consider it essential that the authors clarify the type of questionnaire and the modifications that were made in relation to the original questionnaire.
I suggest introducing in the method a section entitled "Data analysis", to facilitate the reader's understanding of the analyzes performed and the statistical program used.
Line 401-403: Remove this information from the conclusions section and enter it into the discussion.
Bibliography: I suggest the authors review the following bibliographical citations:
Calvente Rejón, I. (2016). Tratamiento de los Juegos Paralímpicos por parte de los medios de comunicación: Ambición y coraje en los Paralímpicos de Río 2016. Shields, N., Synnot, A. J., & Barr, M. (2012). Perceived barriers and facilitators to physical activity for children with disability: a systematic review. Br J Sports Med, 46(14), 989-997. Dowling, M., Brown, P., Legg, D., & Beacom, A. (2018). Living with imperfect comparisons: The challenges and limitations of comparative paralympic sport policy research. Sport management review, 21(2), 101-113. Solves, J., Sánchez, S., & Rius, I. (2018). The prince and the pauper: Journalistic culture and Paralympic Games in the Spanish print press. Journalism, 19(12), 1713-1729.Author Response
Rev,1 B
I am submitting a revised version of my paper entitled „ Polish Paralympic sports in the opinion of athletes and coaches in retrospective studies”. I gave each of the points raised by the reviewers careful consideration and made all changes needed to fulfill their intention as well as formal requirements. The corrections and amendments are marked red in the text.
Response to Reviewer 1 Comments
Comments and Suggestions for Authors
Point 1 In this manuscript, the authors examined “Polish Paralympic sports in the opinion of athletes and coaches in retrospective studies”. To this end, the authors used data from 581 people. As far as I know, the authors did a good job, especially, directing the data analysis. It seems that the results showed support for the objectives and the findings were discussed slightly. In general, I found that this is a well-written article that explores important ideas of relevance to the context of Polish Paralympic Sports. Therefore, I recommend some important revisions and taking into account some modest recommendations that could improve the document:
Line 33: Remove the word "problem", and replace it with a less ambiguous one.
Response 1 “Limitation” has been used instead.
Point 2Line 36-50: There are many influential aspects in Paralympic sport in a very superficial way. The authors express that there are no similar studies in Polish samples, however, they do not reflect similar studies in other countries. It would be necessary and interesting to include studies identifying the main difficulties encountered in sports for people with disabilities, even in different countries. In relation to the comments, I consider that the bibliography is poorly updated. I recommend to the authors the detailed search of more current studies related to the subject of study.
Response2 Changes have been made, new references have been used to improve the text
Point 3Figure 1: Congratulate the authors for the complete description of the sample.
Response3 - Thank you for your appreciation
Point41Line 180: I suggest putting a title that refers to the paragraph that follows, and removing "The process of integration of Polish sports communities" along with the bibliographic reference, because it is confusing if it refers to a title or not.
Response4 This does not refer to the title and has been added to the references - corrected
Point 5Methods: There is no information about the modified questionnaire used, nor the type of questions asked, nor the type of answers. Were they all open questions? Were there closed questions? Were there open and / or closed answers? Likert responses? I consider it essential that the authors clarify the type of questionnaire and the modifications that were made in relation to the original questionnaire.
Response5 The tool was validated and presented in the publication (Sobiecka, J. The image of a Polish Paralympian. University School of Physical Education in Krakow: Kraków, 2013, Monografie No. 22, pp. 49-59 ) [18]. In the presented study, only some of them were used to achieve the study aim.
Point6 I suggest introducing in the method a section entitled "Data analysis", to facilitate the reader's understanding of the analyzes performed and the statistical program used.
Response6 This has been supplemented and explained
Point7Line 401-403: Remove this information from the conclusions section and enter it into the discussion.
Response7 This has been moved to discussion
Point 8 Bibliography: I suggest the authors review the following bibliographical citations:
Calvente Rejón, I. (2016). Tratamiento de los Juegos Paralímpicos por parte de los medios de comunicación: Ambición y coraje en los Paralímpicos de Río 2016.
Shields, N., Synnot, A. J., & Barr, M. (2012). Perceived barriers and facilitators to physical activity for children with disability: a systematic review. Br J Sports Med, 46(14), 989-997.
Dowling, M., Brown, P., Legg, D., & Beacom, A. (2018). Living with imperfect comparisons: The challenges and limitations of comparative paralympic sport policy research. Sport management review, 21(2), 101-113.
Solves, J., Sánchez, S., & Rius, I. (2018). The prince and the pauper: Journalistic culture and Paralympic Games in the Spanish print press. Journalism, 19(12), 1713-1729.
Response8 In accordance with the suggestion of the reviewers and the editor, the most recent references have been added to improve the argumentation of the introduction to the problems discussed in the paper. They have been used in the Discussion section. The following reference items have been added:
Dowlinga M., Brown b , Leggc D, Beacom: A Living with imperfect comparisons: The challenges and limitations of comparative paralympic sport policy research Sport Management Review 21 (2018) 101–113 Castro-Sánchez, M., Zurita-Ortega, F., Chacón-Cuberos, R., López-Gutiérrez, C. J., & Zafra-Santos, E. (2018). Emotional Intelligence, Motivational Climate and Levels of Anxiety in Athletes from Different Categories of Sports: Analysis through Structural Equations. International journal of environmental research and public health, 15(5), 894. doi:10.3390/ijerph15050894 Calvente Rejón, I. (2016). Tratamiento de los Juegos Paralímpicos por parte de los medios de comunicación: Ambición y coraje en los Paralímpicos de Río 2016. Kirk B1, Pugh JN2, Cousins R3, Phillips SM Concussion in University Level Sport: Knowledge and Awareness of Athletes and Coaches.Sports (Basel).2018 Sep 20;6(4). pii: E102. doi: 10.3390/sports6040102. Madden, R. F., Shearer, J., & Parnell, J. A. (2017). Evaluation of Dietary Intakes and Supplement Use in Paralympic Athletes. Nutrients, 9(11), 1266. doi:10.3390/nu9111266 Kwok Ng & Kasper Mäkelä & Jari Parkkari & Lasse Kannas & Tommi Vasankari & Olli J. Heinonen & Kai Savonen & Lauri Alanko & Raija Korpelainen & Harri Selänne & Jari Villberg & Sami Kokko, 2017. "Coaches’ Health Promotion Activity and Substance Use in Youth Sports," Societies, MDPI, Open Access Journal, vol. 7(2), pages 1-11, April Shields, N., Synnot, A. J., & Barr, M. (2012). Perceived barriers and facilitators to physical activity for children with disability: a systematic review. Br J Sports Med, 46(14), 989-997. Solves, J., Sánchez, S., & Rius, I. (2018). The prince and the pauper: Journalistic culture and Paralympic Games in the Spanish print press. Journalism, 19(12), 1713-1729
Round 2
Reviewer 1 Report
In the New Submitted version of the article:
- The introduction part was not supplemented qualitatively;
- The presentation of procedures of respective research remains unclear.
- Line 80 - an uncertain percentage distribution of respondents is presented.
Author Response
1. The introduction part was not supplemented qualitatively;
As suggested by Reviewer 2, the contents introduced already during the first revision enriched the justification for the research, which is focused on the elite Paralympic athletes. In the opinion of the authors of this study, the inclusion of the contents which are undoubtedly very important, discussing the unquestionable importance of sports activity for the process of rehabilitation of people with disabilities, their life satisfaction and the way of their self-realization through amateur sports activity was not the subject of the research presented in the current study. The opinions of professional Paralympic athletes about sports training, perception of themselves as athletes, experience gained and generated expectations are certainly absolutely different from those of amateurs of physical activity and unrelated to the topic and goal set by the authors of this study. Unfortunately, we cannot agree with such changes, in which we would move the fundamental research problem to the background because the topic "Polish Paralympic sports..." indicates that the research is carried out only in an elite group of disabled sportsmen, sportswomen and their coaches.
2. The presentation of procedures of respective research remains unclear.
This chapter has been supplemented and extended with the text, which in part reproduces the visualisation of Fig. 2
3. Line 80 - an uncertain percentage distribution of respondents is presented.
This part has been supplemented and style has been improved
Reviewer 2 Report
The authors have done a good job to improve the article.However, there is one aspect that needs to be reviewed: - The authors refer to the questionnaire they have used
(Sobiecka [25]), however, the authors must include information and
describe the questionnaire used (type of questionnaire, questions,
etc.). Although they have used a questionnaire already validated,
it is very important that the authors describe it. - I cannot access the article that the authors cite in relation
to the questionnaire. I would appreciate it if the authors sent
the questionnaire used. - Depending on the type of the questionnaire it would be necessary
to perform the reliability analyzes if not How do the authors
know that the questionnaire is measuring what they want to
measure in the selected sample?.
In short, they should describe the measuring instrument used to
improve the methodological part of the work.
Author Response
The authors have done a good job to improve the article.
However, there is one aspect that needs to be reviewed: - The authors refer to the questionnaire they have used
(Sobiecka [25]), however, the authors must include information and
describe the questionnaire used (type of questionnaire, questions,
etc.). Although they have used a questionnaire already validated,
it is very important that the authors describe it. - I cannot access the article that the authors cite in relation
to the questionnaire. I would appreciate it if the authors sent
the questionnaire used. - Depending on the type of the questionnaire it would be necessary
to perform the reliability analyzes if not How do the authors
know that the questionnaire is measuring what they want to
measure in the selected sample?.
In short, they should describe the measuring instrument used to
improve the methodological part of the work.
Thank you for your constructive comments. We are convinced that the attached tool along with the description allows for the repeatability of the research. The questionnaire form (chapter of the method) was described and the tool in the English version was attached. as an appendix to the paper
Round 3
Reviewer 1 Report
I would recommend clarifying the statement describing the number of respondents on line 80-82.
Author Response
line 80-81 is corrected,
Reviewer 2 Report
I appreciate that the authors have endeavored to respondto my suggestions. I believe that with the information regarding the
questionnaire used, the article has improved significantly
Author Response
line 80-81 is corrected